# Conduction Disorders after Surgical Aortic Valve Replacement Using a Rapid Deployment Aortic Valve Prosthesis: Medium-Term Follow-Up

**DOI:** 10.3390/jcm12052083

**Published:** 2023-03-06

**Authors:** Christian Mogilansky, Parwis Massoudy, Markus Czesla, Robert Balan

**Affiliations:** Department of Cardiac Surgery, Klinikum Passau, 94032 Passau, Germany

**Keywords:** rapid deployment aortic valve prosthesis, conduction disorders, left bundle branch block (LBBB)

## Abstract

Background: We have previously reported that the incidence of postoperative conduction disorders, especially left bundle branch block (LBBB), after implantation of a rapid deployment Intuity™ Elite aortic valve prosthesis (Edwards Lifesciences, Irvine, CA, USA), was significantly increased compared with conventional aortic valve replacement. We were now interested in how these disorders behaved at intermediate follow-up. Methods: All 87 patients who had undergone surgical aortic valve replacement (SAVR) using the rapid deployment Intuity™ Elite prosthesis and were shown to have conduction disorders at the time of hospital discharge were followed up after surgery. These patients’ ECGs were recorded at least 1 year after surgery, and the persistence of the new postoperative conduction disorders was assessed. Results: At hospital discharge, 48.1% of the patients had developed new postoperative conduction disorders, with LBBB being the predominant conduction disturbance (36.5%). At medium-term follow-up (526 days, standard deviation (SD) = 169.6, standard error (SE) = 19.3 days, respectively), 44% of the new LBBB and 50% of the new right bundle branch block (RBBB) had disappeared. There was no new atrio-ventricular block III (AVB III) that occurred. One new pacemaker (PM) was implanted during follow-up because of AVB II Mobitz type II. Conclusions: At medium-term follow-up after the implantation of a rapid deployment Intuity™ Elite aortic valve prosthesis, the number of new postoperative conduction disorders, especially LBBB, has considerably decreased but remains high. The incidence of postoperative AV block III remained stable.

## 1. Introduction

Rapid deployment aortic valve prostheses are associated with an increased incidence of postoperative conduction disorders compared with conventional aortic valve replacement [1,2,3,4,5,6,7,8,9]. Depending on the prosthesis type (interventional or surgical valves with self-expandable or balloon expandable stents), the distribution of the conduction anomalies can vary considerably. The position of the prosthesis-stent in the left ventricle’s outflow tract is crucial for the development of conduction disorders [10,11,12,13,14].

The population of patients receiving transcatheter aortic valve replacement (TAVR) has already been investigated with regard to not only the incidence of LBBB but also the reduction during follow-up. At 1 year after the implantation, a reduction of 30% to 40% was described [11]. No follow-up data concerning the persistence of conduction disorders after implantation of an Intuity™ prosthesis are available. In general, respective data have not yet been published for the SAVR population. The aim of the present study was to investigate the persistence of postoperative conduction disturbances at medium-term follow-up after implantation of the Intuity™ Elite prosthesis™.

## 2. Materials and Methods

This is a single-center, nonrandomized cohort study following patients who had undergone aortic valve replacement with the rapid-deployment Intuity™ Elite aortic valve prosthesis (Edwards Lifesciences, Irvine, CA, USA), between May 2014 and May 2017. In brief, 200 consecutive all-comer patients were retrospectively evaluated concerning the new incidence of conduction disorders (LBBB, RBBB, and AVB III) after surgery. Surgeries included minimally invasive and conventional approaches, isolated and combined procedures, not excluding endocarditis [1]. Finally, 87 patients had developed new LBBB, RBBB, or AVB III and entered the present study (Table 1).

At least one year after surgery, the patients and their general practitioners and/or cardiologists were contacted. ECGs of the 87 affected patients were collected, and the persistence of the conduction disorders was evaluated. The ECGs were written as twelve-channel surface ECGs. ECGs were written in our hospital or on an outpatient basis by the general practitioner/cardiologist. All ECGs written on an outpatient basis were sent to the department of cardiac surgery at Klinikum Passau and evaluated here for the presence of any conduction abnormality. Among the patients who had received a pacing device because of a new postoperative AVB III, the pacing rate of the implanted device was evaluated. Devices included pacemakers, implantable cardioverter defibrillators (ICD), and cardiac resynchronization therapy- defibrillators (CRT-D).

### Data Acquisition and Statistics

Data are descriptively presented as means with standard deviation, standard error of the mean for continuous variables and, if applicable, as percentages for categorical variables.

## 3. Results

### 3.1. Perioperative Data

Perioperative data had been reported previously [1]. In brief, the mean patient age was 71 years, with 39% of patients being female. Twenty-seven patients had a history of stroke, and nine had aortic valve endocarditis. The mean ejection fraction was in the low normal range, and 164 patients were in New York Heart Association (NYHA) class III or IV. The mortality rate of 3.3% had been in accordance with the 3.8% predicted by EuroSCORE II. Concerning the performance of the rapid-deployment aortic valve prosthesis, the mean pressure gradient had been little more than 8 mmHg, and only 2 of 183 patients had a paravalvular leak (PVL) that was more than trivial.

The incidence of new postoperative conduction disorders was 48.1%. LBBB was the most frequent conduction disturbance (36.5%). RBBB occurred in 2.3% and AVB III in 9.3%. All patients with AV-Block III had been treated with PM, ICD, or CRT-D (Table 1).

### 3.2. Medium-Term Follow-Up

A total of 87 patients with postoperative conduction disorders were followed up for at least one year after surgery. Data were complete for 78 patients (59 LBBB, 4 RBBB, and 15 AVB III), representing a follow-up rate of 90%. Nine patients were lost to follow-up; they could not be contacted either by telephone or by mail. In addition to that, in all these nine cases, neither general practitioners nor cardiologists had any follow-up information. The mean time between surgery and the follow-up ECG was 19 months (12 to 34 months).

At the time of follow-up, LBBB had disappeared in twenty-six patients (44%) and RBBB in two patients (50%). Among the patients who had received a pacing device after surgery, the pacing rate of the implanted device (PM, ICD, or CRT-D) was evaluated. Nine PM patients (60%) showed to have a pacing rate over 99%, three patients (20%) had a pacing rate between 56%, and 99% and one patient (6.7%) had a pacing rate of 0.2%. Two of the patients who had received a pacemaker after surgery were lost to follow-up. Patients with an ICD or CRT-D had a device activity of 100%. No new cases of AVB III occurred during follow-up. One patient with LBBB developed an AV block II Mobitz type and received a new pacemaker during follow-up (Table 2).

## 4. Discussion

Conduction abnormalities after aortic valve implantation are a well-known incident and more often occur after SAVR with a sutureless aortic valve prosthesis than after SAVR with a conventional prosthesis [3,5,6,8,9,15,16]. Among the conduction abnormalities observed, the incidence of LBBB after conventional SAVR ranges from 1.5% to 6% [4,9,17]. In contrast, the incidence of LBBB after SAVR with the Intuity™ prosthesis is much higher, with reported values between 30.2% and 34.1% [9,14,18,19,20]. In our previous work, we had even found an incidence of 36.5%, with LBBB being much more frequent than any other conduction abnormality [1].

Physiologically speaking, LBBB represents a loss of ventricles’ contraction synchrony and is a serious perioperative complication, especially in young patients. In the current literature, LBBB is clearly associated with worsening of left ventricular function and, lately, even with an increase in mortality during follow-up in affected patients [11,21,22,23,24,25]. However, an encouraging aspect is that almost 60% of patients with LBBB after TAVR had a normal electrocardiogram at 1-year follow-up (11).

Our present work is the first to focus on these data in a SAVR population. In parallel to the TAVR population, we observed a decrease in LBBB at medium-term follow-up, at least 1 year after surgery. However, LBBB remained persistent in 56% of the patients who had acquired it after surgery and only regressed in 44%. The rate of RBBB decreased by one half, and none of the LBBB and RBBB progressed to AVB III during follow-up. In patients with postoperative AVB III and subsequent device implantation (PM, ICD, CRT-D), AV block III completely regressed in only one patient (pace-rate < 1%), changed to intermittent block in three patients (pace-rate 56.4%, 75.9%, and 87.6%) and was persistent in eleven patients (pacing-rate > 99%). The persistence of the AV block III in the patient who had received a CRT-D device could not be assessed by pace-rate alone, because of permanent resynchronization pacing.

Immediately postoperatively, the Intuity™ Elite prosthesis, had shown excellent hemodynamic properties and low rates of PVL on the one hand but high rates of postoperative conduction disorders on the other hand. These data had caused a certain reluctance to implant the prosthesis, especially in younger patients, and had motivated us to perform the study presented herein. The decreasing trend at medium-term follow-up is certainly encouraging. However, the remaining 19% of the overall population, who continue to have left bundle brunch block more than one and a half year after surgery deserve further follow-up, including information on functional and clinical data.

In addition to age, frailty, patient-prosthesis mismatch, paravalvular leakage and need for pacemaker implantation, the incidence and persistence of left bundle brunch block thus deserves consideration when the heart team tries to find the best prosthesis for any individual patient with aortic stenosis.

Our study has limitations. This is a single-center cohort study carrying all the disadvantages of non-randomized studies, but on the other hand, offers the advantages of a real-world registry. Nine patients could, unfortunately, not be contacted, only offering a 90% complete follow-up.

## 5. Conclusions

We conclude that, after SAVR with the rapid deployment Intuity™ Elite aortic valve prosthesis, the number of postoperative conduction disorders considerably decreases at medium-term follow-up. However, the percentage of left bundle branches, being the most frequently observed conduction abnormality, remained elevated at 19% of the original population.

The present results confirm our policy of reluctance to liberally implant this rapid deployment prosthesis because we observed a persistence of left bundle branch block in almost 20% of our patients at medium-term follow-up. Future investigation is warranted to further follow the fate of LBBB and to monitor the hemodynamic and clinical data of the respective patients.

In the meantime, the incidence and persistence of left bundle branch block after the implantation of any specific aortic valve prosthesis should be considered when the heart team decides upon the best treatment for a patient with aortic valve stenosis.

## Figures and Tables

**Table 1 jcm-12-02083-t001:** Postoperative conduction disorders.

LBBB	66/181 (36.5)
RBBB	4/176 (2.3)
AV block III	17/183 (9.3)
PM implantation	15/183 (8.2)
ICD implantation	1/183 (0.5)
CRT-D implantation	1/183 (0.5)

Values are given as absolute numbers with % in parentheses. AV block III, atrioventricular block III; CRT-D, cardiac resynchronization therapy–defibrillator; ICD, implantable cardioverter defibrillator; LBBB, left bundle branch block; RBBB, right bundle branch block.

**Table 2 jcm-12-02083-t002:** Postoperative conduction disorders at medium-term follow-up (526 days, SD = 169.6, SE = 19.3).

Persistent LBBB	33/59 (55.9)
Persistent RBBB	2/4 (50)
PM activity < 1%	1/15 (6.7)
PM activity 56–99%	3/15 (20)
PM activity > 99%	9/15 (60)
ICD activity > 99%	1/1 (100)
CRT-D activity > 99%	1/1 (100)
PM after hospital discharge ^1^	1/1 (1.4)

Values are given as absolute numbers with % in parentheses. AV block III, atrioventricular block III; CRT-D, cardiac resynchronization therapy–defibrillator; ICD, implantable cardioverter defibrillator; LBBB, left bundle branch block; RBBB, right bundle branch block. ^1^ Patient with LBBB and AV block II Mobitz type II.

## Data Availability

Not applicable.

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
