# Peer review of "Conduction Disorders after Surgical Aortic Valve Replacement Using a Rapid Deployment Aortic Valve Prosthesis: Medium-Term Follow-Up"

_jcm, 2023, doi:10.3390/jcm12052083_

Round 1

Reviewer 1 Report

Could, you, please show any data about preoperative computed tomographic (CT) images, oversizing/undersizing, and the number of postoperative conduction disorders?

Eur J Cardiothorac Surg . 2022 Mar 24;61(4):899-907.  doi: 10.1093/ejcts/ezab431.

3-Dimensional computed tomographic assessment predicts conduction block and paravalvular leakage after rapid-deployment aortic valve replacement

Sung Jun Park 1Younju Rhee 2Chee-Hoon Lee 3Ho Jin Kim 4Joon Bum Kim 4Suk Jung Choo 4Jae Won Lee 4 Affiliations expand

  • PMID: 34687534
  •  
  • DOI: 10.1093/ejcts/ezab431

Abstract

Objectives: Complications like complete atrioventricular block (CAVB) and paravalvular leakage (PVL) following rapid deployment aortic valve (AV) replacement (RDAVR) remain unresolved. Selecting an optimal size of the valve might be important to minimize the incidence of these complications. We sought to determine the impact of prosthesis size relative to the anatomic profile of the AV on the occurrence of CAVB or PVL after RDAVR.

Methods: Preoperative computed tomographic (CT) images were evaluated in patients receiving RDAVR (INTUITY ELITE) between February 2016 and December 2019. The occurrence of CAVB requiring permanent pacemaker implantation and PVL (≥ mild) was evaluated. The relative size of implants against the cross-sectional dimensions of recipients' AV annulus and left ventricular outflow tract (LVOT) were calculated.

Results: Among 187 eligible patients, CAVB and PVL (≥ mild) occurred in 12 (6.4%) and 11 patients (5.9%), respectively. CAVB was associated with oversized RDAVR (RDAVR frame width minus average diameter of LVOT calculated from the cross-sectional area [ΔLVOTarea]: odds ratio, 2.05; 95% confidence interval, 1.28-3.30): this was with an area under the curve of 0.78 (P = 0.005). The projected probability of CAVB was <3% when the ΔLVOTarea was <1.3. In contrast, PVL was associated with under-sized RDAVR (RDAVR size divided by the longest diameter of AV annulus [index Annlong]: odds ratio, 0.64; 95% confidence interval, 0.51-0.79): This was with an area under curve of 0.94 (P < 0.001).

Conclusions: CT parameters of the AV annulus and LVOT are highly reliable in the prediction of CAVB or PVL after RDAVR. Our data might justify CT-based sizing of prosthesis for RDAVR.

Keywords: Complete atrio-ventricular block; Computed tomography; Left ventricular outflow tract; Paravalvular leakage; Rapid deployment aortic valve replacement.

Author Response

Could, you, please show any data about preoperative computed tomographic (CT) images, oversizing/undersizing, and the number of postoperative conduction disorders?

In our institution we perform before each isolated aortic valve replacement a CT Thorax scan to analyse if a minimally invasive approach is feasable. This is a native CT scan which is not suitable for analysing the size of the aortic annulus.

The present manuscript is a follow up study of a publication we presented in 2018, reporting the same cohort of patients (Mogilansky et al, ICVTS 2018). The exact number of conduction disturbances is presented in Table1.

Have you, ever, read a paper about this topic whose abstract is down written?

The article mentioned above is interesting suggesting important data regarding the ct scan based sizing of aortic valves.     

We had a specific training regarding the correct sizing of the Intuity valves by Edwards Lifesciences, in order to prevent oversizing or undersizing. Undersizing would lead to paravalvular leakages and oversizing may lead to rupture of the annulus.

Is there any link between PVL and postoperative conduction disorders in your study?

We only had one case of PVL so that a statistic work up was not meaningful.

Reviewer 2 Report

Great observation of the sutureless valve by Edwards. I would like the authors to also comment on the patient's ECHO findings post-op at the midterm time point as they point to higher morbidity and LV function in patients with LBBB. I look forward to seeing more outcomes from this study.

Author Response

Great observation of the sutureless valve by Edwards. I would like the authors to also comment on the patient`s ECHO findings post-op at the midterm time point as they point to higher morbidity and LV function in patients with LBBB. I look forward to seeing more outcomes from this study.

Thank you very much for your positive reaction. The reviewer points out an important question regarding the follow up echoes of this cohort of patients. The focus of the study submitted here was on the persistence at midterm of conduction discordances after aortic valve replacement with an Intuity prosthesis. Functional tests or echocardiographic scans were not included but are planned for the future.